# Thermo-Visco-Elastometry of RF-Wave-Heated and Ablated Flesh Tissues Containing Au Nanoparticles

**DOI:** 10.3390/bios13010008

**Published:** 2022-12-22

**Authors:** Bayan Kurbanova, Zhannat Ashikbayeva, Aida Amantayeva, Akbota Sametova, Wilfried Blanc, Abduzhappar Gaipov, Daniele Tosi, Zhandos Utegulov

**Affiliations:** 1Department of Physics, School of Sciences and Humanities, Nazarbayev University, Astana 010000, Kazakhstan; 2School of Engineering and Digital Sciences, Nazarbayev University, Astana 010000, Kazakhstan; 3Université Côte d’Azur, INPHYNI, CNRS UMR7010, Avenue Joseph Vallot, 06108 Nice, France; 4Department of Medicine, Nazarbayev University School of Medicine, Astana 010000, Kazakhstan; 5National Laboratory Astana, Laboratory of Biosensors and Bioinstruments, Astana 010000, Kazakhstan

**Keywords:** radiofrequency, heating, hyperthermia, ablation, fiber-optic, sensors, Brillouin spectroscopy, viscoelastic, gold nanoparticles, thermometry, temperature, protein denaturation, flesh, metal nanoparticle, theranostics

## Abstract

We report non-contact laser-based Brillouin light-scattering (BLS) spectroscopy measurements of the viscoelastic properties of hyperthermally radiofrequency (RF)-heated and ablated bovine liver and chicken flesh tissues with embedded gold nanoparticles (AuNPs). The spatial lateral profile of the local surface temperature in the flesh samples during their hyperthermia was measured through optical backscattering reflectometry (OBR) using Mg–silica-NP-doped sensing fibers distributed with an RF applicator and correlated with viscoelastic variations in heat-affected and ablated tissues. Substantial changes in the tissue stiffness after heating and ablation were directly related to their heat-induced structural modifications. The main proteins responsible for muscle elasticity were denatured and irreversibly aggregated during the RF ablation. At T > 100 °C, the proteins constituting the flesh further shrank and became disorganized, leading to substantial plastic deformation of biotissues. Their uniform destruction with larger thermal lesions and a more viscoelastic network was attained via AuNP-mediated RF hyperthermal ablation. The results demonstrated here pave the way for simultaneous real-time *hybrid optical sensing of viscoelasticity and local temperature* in biotissues during their denaturation and gelation during hyperthermia for future applications that involve mechanical- and thermal-property-controlled theranostics.

## 1. Introduction

For localized treatment of cancer in the liver, brain, and breast [1,2] and of Barrett’s esophagus syndrome, varicose veins, and uterine fibroids, *radiofrequency ablation (RFA)* at mild powers is often employed as a minimally invasive, safe, and low-risk therapeutic procedure [3,4]. During RFA, the electromagnetic (EM) waves passing through biological tissues via an applied needle cause their local heating due to the propagation of an alternating electrical current (AC) modulated at frequencies ranging from 100 kHz to 300 GHz [5], which causes ionic agitation and cellular necrosis, which can ultimately cause tissue coagulation [6]. However, a lack of control of thermal energy deposition during RFA can lead to such undesirable therapeutic results as malignant transformation [7]. Therefore, the control of the RFA procedure is essential for defining how much tissue is destroyed while avoiding charring and excessive damage to surrounding healthy tissues.

Compared to other EM wave sources, such as visible and near-infrared light, RFA causes much deeper heat penetration into tissues and, therefore, is applicable for the treatment of deeply localized solid tumors [8]. However, it is impossible to generate desirable ablation zones that cover spatially extended tumors, and the employment of a large RF power and a longer time in order to extend the ablation zone will lead to undesirable tissue burns. Adding a mediating absorptive agent, such as *metal nanoparticles (NPs)*, expands the ablation zone during RFA. Hyperthermal heating occurs in the nanometer-scale vicinity of embedded NPs, thus increasing the local temperatures up to hundreds of degrees on a micrometer scale, leading to a more spatially extended temperature distribution within the heated sample, which causes the disruption of cells in the vicinity. More importantly, loading selected NPs with high thermal and electrical conductivities driven by free electrons into tissues promotes wider and deeper ablation zones due to random spatial distributions of localized heating sources attributed to these metal NPs [9].

Gold nanoparticles (AuNPs) are promising for the targeted thermal destruction of cancer due to their abilities of surface functionalization and hyperthermal RF heating. Moreover, AuNPs are less toxic than other metallic NPs and do not chemically interact with most biological substances [10]. The multifunctionality of AuNPs makes them ideal candidates for theranostic applications [11]. Furthermore, with the increase in NP concentration, the ablation efficiency increases. However, NPs with sizes smaller than 5 nm and larger than 100 nm are not biologically active. Very small NPs are easily cleared by the kidney, while NPs larger than 100 nm are sequestered by the white blood cells that defend the immune system [12].

The viscoelastic properties of tissues vary due to tissue dehydration, protein denaturation, and coagulation during hyperthermal treatments, and these properties make them good candidates for dosimetry [13]. Monitoring *viscoelastic variations* in tissues is critical during thermal ablation, since cell death under hyperthermal treatment is driven by AC-induced thermal stresses in the presence of magnetic nanofluids [14]. It was revealed that thermal conduction and tissue volume relaxation parameters play a significant role in tissue bio-thermo-mechanics [15]. A fully nonlinear thermo-visco-hyperelastic finite-element algorithm was developed to describe the nonlinear characteristics of bioheat transfer in deformed soft tissues with thermal expansion/shrinkage during hyperthermal treatment [16].

Turning to the effects of AuNPs on the elastic properties of tissue, the conjugation of AuNPs to fibrilized collagen, as a soft tissue filler, leads to the formulation of AuNP–collagen constructions and increases the collagen’s crosslinking longevity. The mechanical properties of tissues are enhanced as a function of crosslinking [17]. However, to the best of our knowledge, no investigations have been performed on the *thermo-viscoelasticity of tissues during and after AuNP-mediated hyperthermal* treatment. Additionally, the viscoelasticity monitoring of RF-heated tissues is very critical if one wants to understand and control the evolution of the mechanical properties of tissues during and after RF heating and ablation. In addition, it is well known that the mechanical properties of the extracellular matrices of biological tissues and their biopolymer components are essential for normal tissue function, and disturbances in these properties frequently occur in disease [18].

The awareness that diseases ranging from osteoarthritis to atherosclerosis, cancer, and diabetes are linked to alterations in the mechanical properties of the affected tissue has stimulated research on the mechanical properties of entire tissues [18]. Since proteins are major components of flesh (e.g., beef or chicken), it is important to emphasize that the *viscoelastic properties of RF-heated and ablated flesh will be dominated by the viscoelasticity of the major RF-heated and ablated proteins* constituting these flesh tissues. In terms of the percentage of mass content, chicken flesh actually has more protein than beef flesh [19]. Heat-ablated tissue is stiffer than untreated tissue and is used to measure spatial stiffness variations and to quantify Young’s modulus in tissues after thermal treatment [20]. The elastic and viscous components of a gelated myofibrillar protein network started to increase from 30 to 45 °C [21]. The formation of a viscoelastic gel network in ribbonfish meat was observed during heating, and the maximum rate of increase in the storage and loss of elastic moduli was found to be in the temperature range of 56.8 to 63.3 °C [22]. However, the viscoelastic properties of beef muscle showed a slight decrease for T < 55 °C, then a sharp increase for 55 °C < T < 80 °C, and a saturation at higher temperatures [23]. During the heating of cancerous biological tissues, the cytotoxicity of tumors occurred at 42 °C, while protein coagulation started at 60 °C. The temperature range of 60 to 100 °C is known as an effective parameter for obtaining the largest ablation area. However, temperatures above 100 °C cause tissue evaporation and carbonization, which have undesirable effects [24].

For the assessment of the elastic and viscous properties of tissues, various techniques have been utilized, such as ultrasound elastography [25], dynamic nanoindentation [20], dynamic rheology [21], and Brillouin light-scattering (BLS) spectroscopy [26]. The latter technique is based on the inelastically scattered laser light from thermally excited GHz acoustic waves in a probed condensed medium. BLS spectroscopy is a unique non-destructive and non-contact technique for probing the viscoelastic properties of various materials, and it was traditionally applied in soft matter and materials science studies [27,28], geosciences [29], biomedical applications [30,31], and the detection of biomarkers [32,33]. More importantly, advances in confocal micro-Brillouin light scattering have enabled the rapid noninvasive monitoring of tissue biomechanics down to the micrometer scale, thus opening the way for live cell imaging [34,35]. Turning to cancer detection, Brillouin shifts in non-regressing and regressing melanomas and in healthy tissue samples were found significantly different, with the healthy tissue being the softest and non-regressing melanoma being the stiffest [36]. The elastic moduli of muscle and rat tail tendon collagen were obtained, and hydrogen bond force constants were estimated with BLS [37,38]. In addition, BLS allowed the observation of the propagation of sound waves along a one-dimensional periodic array of sarcomeres in rabbit psoas muscle myofibrils [39]. A biomechanical contrast of muscular, connective, epithelial, and nervous tissues was presented through high-resolution Brillouin microscopy imaging [26]. A temperature-dependent stiffening and gelation process of albumin from chicken eggs was reported through BLS [40].

As for local optical thermometry, the OBR technique stands as the most modern approach to temperature sensing with fiber-optic sensors (FOS) during thermal ablation. In particular, Mg–silicate-NP-doped fibers have greater backscattering signals than those of standard single-mode fibers, thus allowing the extension of OBR to the real-time measurement of the spatial distribution of temperature over the inner plane of thermal ablation, providing a valuable alternative to thermal imaging with a simpler implementation [41]. Thermal modifications in photodamaged biological samples can be experimentally monitored by chaotic attractors, which depend on optical transmittance [42].

In this work, we report local viscoelastic variations through BLS spectroscopy within chicken muscle and bovine liver tissues with embedded AuNPs *after* RF heating and ablation. The distributions of Mg–silicate-NP-doped sensing fibers along an RF applicator during ablation allowed for the planar measurement of the local temperature profile and assessment of the ex vivo RF-ablated zone. BLS combined with Mg–silicate-NP-doped sensing fibers allowed us to obtain *viscoelastic sensing correlated with the detection of the local temperature* of AuNP-mediated hyperthermally RF-heated and ablated flesh tissues. The nature of the high Brillouin peak shift and broadening contrast between normal tissue and the created thermal lesions was studied. In addition, the relationship between the change in viscoelasticity and the obtained thermal dose of the tissues was deeply investigated to better understand the AuNP-assisted hyperthermal RF ablation phenomenon.

## 2. Materials and Methods

Brillouin spectra were recorded in the 180°-backscattering configuration by using a 6-pass tandem Fabry–Perot Interferometer TFP-2 (Table Stable Ltd., Mettmenstetten, Switzerland) [43]. The free spectral range was set to 25 GHz. Laser light with a 532 nm wavelength from Verdi-G2 (Coherent, Santa Clara, CA, USA) was used with a mild incident beam power kept below 10 mW to prevent sample damage. The laser spot diameter with a 5× microscope objective was 5 μm. The Brillouin spectral changes were measured at different spatial points on RF-heated and ablated mammal tissues corresponding to the temperatures measured with temperature-sensing fibers.

The ex vivo ablation of fresh chicken and bovine liver tissues was conducted by using an RF/MW Hybrid generator (Leanfa s.r.1.) according to the method in [44], and this was aligned with the European Union’s “Three Rs” principle. Before use, the meat and livers were stabilized to room temperature at 20–21 °C. The RF power for the ablation procedure was set to 60 W, while the frequency value was 450 kHz. To control the temperature change during radiofrequency ablation, four Mg–silicate-NP-doped fibers spliced to single-mode optical fibers (SMFs) were placed in parallel on the y-axis on meat at a 5 mm distance from each other, as depicted in Figure 1. The generator was set to the safe mode, allowing the termination of the ablation procedure when the impedance of 800 Ω was reached. The active electrode (AE), which was in the form of a research-grade single-tip applicator with a 3 mm diameter and a 160 mm length, was inserted between the 2nd and 3rd fibers to deliver the RF waves, while the meat was positioned on a metallic plate that was used as a passive electrode (PE). The positions of both electrodes and sensing fibers were fixed for all of the experiments. A schematic representation of the experimental setup containing the RF generator, Fabry–Perot interferometer, optical backscatter reflectometer (OBR), and the distributed temperature-sensing optical fiber system is shown in Figure 1.

The synthesis of AuNPs was conducted by using the citrate reduction method, as presented by Turkevic et al. [45]. The average size (d = 20 nm) of the NPs was estimated with a transmission electron microscope (TEM, JEOL JEM—1400 Plus, Indianapolis, IN, USA) image, as shown in Figure 2e. Based on previous studies, Au NPs with a size of >20 nm could take a longer time to be excreted from the body during in vivo experiments, as described in [46]. The optimal density of NPs was selected as 1 mg/mL based on outcomes from previous works [47]. The RF ablation was performed in two conditions: on pristine tissues and on tissues treated with AuNPs with a size of 20 nm dispersed in 0.2% agarose solution. A total of 200 µL of AuNPs dispersed in an agarose solution were injected into the tissue by using a syringe in proximity to the active electrode. The ablation was performed three times under the same ablation conditions to measure the post-heating Brillouin spectral variations with a coupled confocal microscope.

The temperature-sensing setup was based on a commercial OBR (Luna Inc., OBR4600, Roanoke, VA, USA) that worked in the continuously distributed sensing mode. The OBR instrument was connected to the sensing fibers with different lengths by 1 × 8 wideband splitter operating in the third optical window. The OBR delivered a signal via a fiber link and then measured the Rayleigh-scattering-induced reflection. Return losses experienced along the fiber length could be estimated by recording the propagation time of backscattered light. The RFA wavelength of each fiber section’s reflection spectrum shifted depending on the change in the temperature. The measurements were taken at 45 mm intervals at the tip of each fiber’s spectrum, and these points were grouped into a 2D matrix based on the physical arrangement of the fibers. The obtained 2D thermal maps in the lateral planes of the tissues were arranged in different colors that indicated the temperature changes. The choice of sensing parameters represented a trade-off between the spatial resolution and temperature accuracy; in the experiments, the spatial resolution was set to 2.0 mm, while the overall ‘useful’ sensing length was set to 45 mm while taking all four fibers into account in this window. All optical connections, including the led-in fibers, splitters, and extenders, were based on a standard single-mode fiber (SMF, corning SMF-28), while the sensing fibers were based on Mg–silicate-NP-doped fibers and had a high spatial resolution (2 mm).

## 3. Results and Discussion

Generally, both chicken and bovine liver flesh consists of approximately 71–75% moisture, 16–17% protein, 5–8% fat, and 1% ash [19]. Chicken muscle tissue contains muscle fibers and connective tissue, where connective tissue covers the muscle fibers, as can be seen in Figure 2a,b. The *myofibrillar proteins* constitute 50 to 55% of the chicken muscle’s total protein content, while *sarcoplasmic proteins* account for approximately 30–34%. The remaining 10–15% of the proteins are *connective tissue proteins*. Myosin and actin are the most important proteins that constitute the myofibrillar structure [48]. The bovine liver is composed of smaller histological structures called lobules, which are roughly hexagonal in shape. In addition, the bovine liver is covered with a connective tissue capsule, which is a thin but tough fibrous supporting connective framework of densely interwoven collagen fibers [49]. The fibers of connective tissue consist of collagen, elastic, and reticular fibers; they are, thus, responsible for the liver’s elasticity and tensibility [50].

The heating process had a considerable effect on the biological tissues’ properties, as the changes in toughness with denaturation temperatures or indirect interpretations of structural changes in tissue components occurred [51]. Figure 3 demonstrates the chicken muscle’s 2D thermal map in the x-y plane, which was processed from the data obtained along the lengths of all four Mg–silicate-NP-doped sensing fibers. The thermal maps of the tissues present the changes in the temperature during the thermal ablation procedure due to the dissipation of RF power in the bovine liver and chicken muscle tissues. The active electrode was positioned at 3.5 cm from the left side, while the sensing system was placed on the right side. The heat pattern clearly showed the maximum temperature increase at the tip of the active electrode and the heat distribution within the tissue, which are depicted in different colors. Images of the RF-ablated bovine liver and chicken meat are shown in Figure 2c,d. Adding AuNPs to the tissues led to a more extensive spatial temperature distribution with a lower maximum temperature; thus, much more uniform temperature distributions along the tissue could be obtained with AuNPs. Hyperthermal RF heating occurred, whereas the AuNPs absorbed the RF waves and produced heat via the limited movement of free electrons inside the NPs. The asymmetric heat pattern was likely due to the heterogeneous properties of the tissue, which deviated from the RF ablation pattern, particularly for fast ablation phenomena.

Figure 4 demonstrates the quantification of the (a) cytotoxic (temperatures > 42 °C) and (b) thermally damaged (temperatures > 60 °C) regions of the chicken muscle tissue for each experimental condition. As can be seen from the bar charts, the AuNP-mediated RFA achieved a larger area of thermal heating and ablation results than it provided in the pristine chicken meat. The same trend was observed for the bovine liver, and both tissues had similar thermal properties.

Generally, chicken meat was found to enhance its stiffness upon heating in two phases. The first phase took place at T = 40–60 °C, which was likely due to the heat-induced denaturation of myofibrillar proteins, especially myosin. The second stage led to further stiffening at T = 65–80 °C, which could be ascribed to the denaturation of intramuscular collagen, which was associated with the initial breakage of hydrogen bonds, thus loosening the fibrillar structure with the subsequent contraction and dissolution of the collagen molecules, as well as their eventual gelation [52]. Most of the water in living muscle is held within the myofibrils, and longitudinal shrinkage of the myofibrils at high temperatures causes great water loss. Other studies suggested that the *thermal injury threshold* for RFA in chicken tissue is 65 °C [53]. After thermal treatment at temperatures in the range from 53 to 63 °C, the bovine liver’s collagen and reticular fibers were denatured, and this led to the breaking of hydrogen bonds and an irreversible transformation of the crosslinked triple-helical structure into a more random coiled structure [54]. At T> 63 °C, the cells had shrunk as much as they could, and most of the water in the cells was forced out [55].

Figure 5a demonstrates the typical Brillouin spectra of pristine and AuNPs-treated chicken muscle tissue corresponding to the measured local temperatures. The back-scattering configuration employed in our BLS measurements enabled us to measure only the bulk longitudinal acoustic waves. The Brillouin peaks decayed in intensity with the increase in the RF-induced temperature due to the heat-induced removal of water content from the tissues, thus effectively reducing the elasto-optic coupling. The ablation temperatures correlated well with the Brillouin shift and linewidth changes, with a frequency contrast from 8 and 1.5 GHz, respectively, at the outer transition zone’s boundary and of 14.3 and 9 GHz at the condensation boundary. The rate of the Brillouin shift and linewidth initially increased sharply and changed slightly at the condensation boundary. At high temperatures from 65 to 100 °C, low mechanical contrast was observed, as most of the cells were destroyed due to coagulative necrosis taking place in this zone. Similar results were obtained by using shear wave elastography methods [48] in which the velocities and attenuation of propagated shear acoustic waves were measured [56].

Figure 5b illustrates the temperature-dependent Brillouin shift (a) and linewidth (b) of pristine and AuNP-treated bovine liver and chicken muscle tissues. Initially, at T = 20–30 °C, the Brillouin shift and linewidth stayed the same, i.e., the tissues did not undergo denaturation, and the mechanical changes were typically reversible for this temperature elevation range. However, at *T* ≥ 40 °C, a slight increase in the Brillouin shift was noticed, which corresponded to the cytotoxic region. A sharp increase in stiffness was observed with further temperature elevation until the onset of ablation. Once ablation occurred, almost constant values of the Brillouin frequency were registered at *T* > 60 °C. The viscous property showed a generally increasing trend as a function of the temperature for both tissues before and after adding AuNPs, as evidenced by the variations in the Brillouin width.

At the initial temperatures, the differences in the Brillouin frequency and linewidth between the muscle and liver tissues were almost 1 GHz. The bovine liver was stiffer and more viscous than the chicken muscle tissue. However, the heating of the pristine tissues up to 110 °C for 40–50 s led to incomplete cell destruction, which was observed in the split Brillouin peaks. The position of the first peak corresponded to the tissue’s initial Brillouin frequency value and was related to the non-damaged tissues, while the second peak shifted to a much higher Brillouin frequency of 15 GHz, corresponding to the likely formation of an organic polymeric phase related to completely burned (dead) tissues. Furthermore, the AuNP-mediated hyperthermal RF ablation even enabled the uniform destruction of tissues at the micrometer scale, where no Brillouin peak splitting took place. The AuNPs with a size of 20 nm absorbed RF energy and quickly released heat into the surrounding region due to the increased electron-surface scattering, since the size of the NPs was significantly smaller than the mean free path of the electrons in gold. As can be seen from Figure 5b, the tissues with AuNPs were stiffer and more viscous compared to the pristine ones. The implementation of metallic AuNPs in soft tissue led to the formation of an NP–biotissue composite medium, and further heating of this metal–organic medium caused the formation of a more viscoelastic gel network with NPs inside through protein denaturation.

## 4. Conclusions

We investigated viscoelastic properties and correlated them with the temperature profiles of AuNP-assisted hyperthermally RF-heated and ablated tissues through confocal Brillouin micro-spectroscopy combined with Mg–silicate-NP-doped temperature-sensing fibers. It was revealed that the significant spectral changes in the Brillouin peak position and linewidth were directly related to the obtained thermal dose and the consequent protein denaturation processes. Increased thermal lesion areas with uniform and complete cell destruction were obtained via the loading of Au NPs into tissues. In addition, compared to pristine tissues without any NPs, further heating of the NP–tissue composite medium led to an increase in the composite tissues’ viscoelastic properties at the same temperatures. These results suggest that our hybrid Brillouin–OBR technique can possibly define a tissue’s structural deformations during thermal therapies, thus encouraging the use of this technique for the dosimetric control of hyperthermia in future applications. This study opens an avenue for real-time simultaneous localized monitoring of the viscoelastic and thermal properties of metal-nanoparticle-embedded bio-tissues across their heat-driven structural phase transitions in hyperthermal and theranostic applications.

## Figures and Tables

**Figure 1 biosensors-13-00008-f001:**
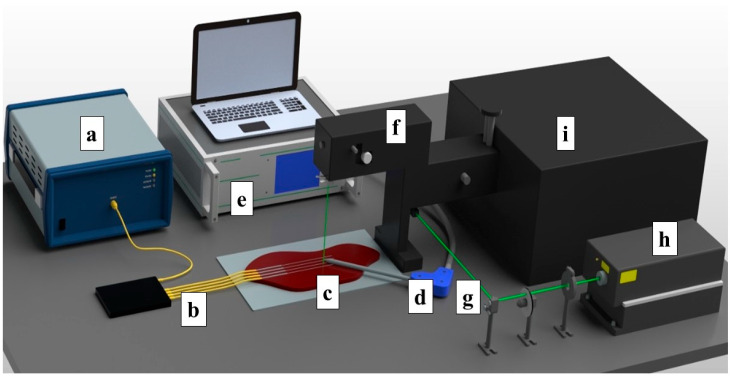
Schematic view of the radiofrequency ablation setup. The setup consisted of: (a) OBR Luna 4600 with a computer used for data acquisition, (b) optical fibers, (c) a bovine liver, (d) the RFA applicator, (e) a hybrid RF/MWA generator used in the RF mode, (f) a confocal microscope, (g) a laser path, (h) a Verdi-G2 532 nm laser source, and (i) a tandem Fabry–Perot interferometer TFP-2.

**Figure 2 biosensors-13-00008-f002:**
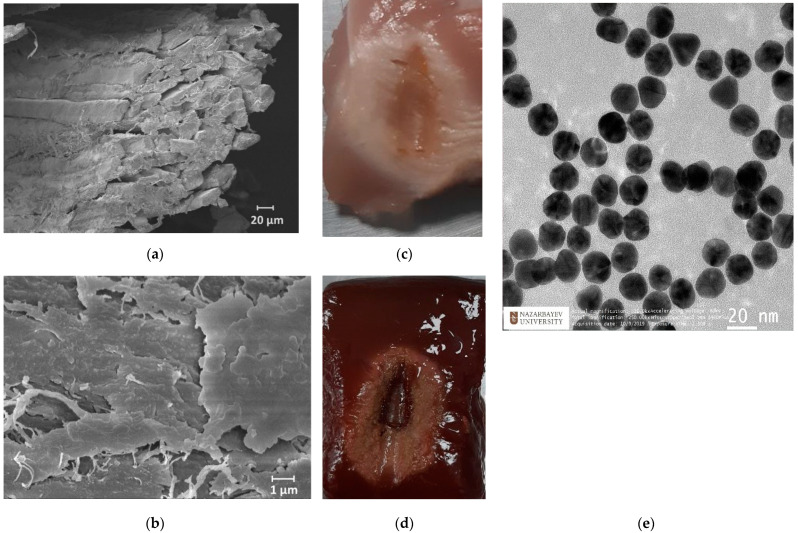
Scanning electron microscopy images of skeletal muscle tissue (**a**) and a closer view of the connective tissues (**b**). Ablated chicken muscle (**c**) and bovine liver (**d**). Transmission electron microscopy image of AuNPs (**e**).

**Figure 3 biosensors-13-00008-f003:**
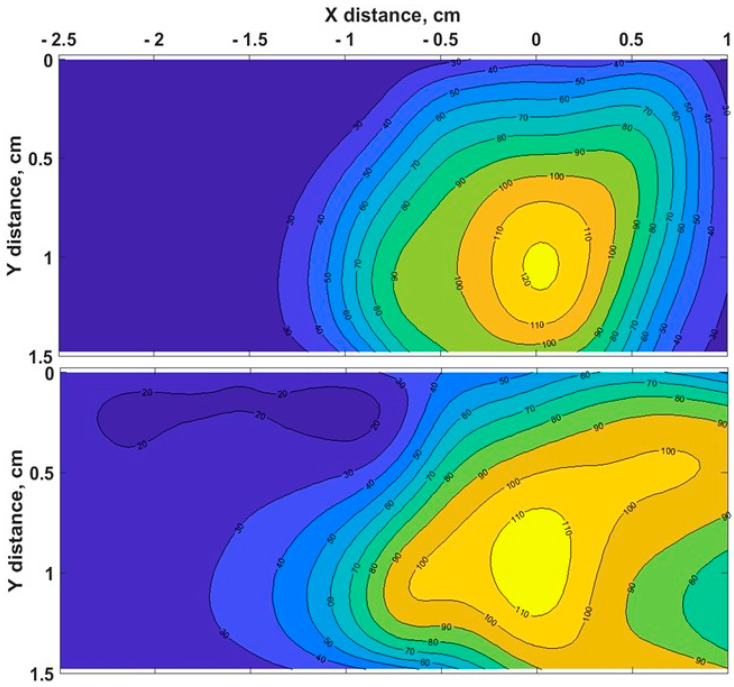
Typical two - dimensional thermal map of pristine chicken muscle (top) and tissues with AuNPs added (down) in the xy lateral plane (x = direction parallel to the RFA applicator and to the sensing fibers).

**Figure 4 biosensors-13-00008-f004:**
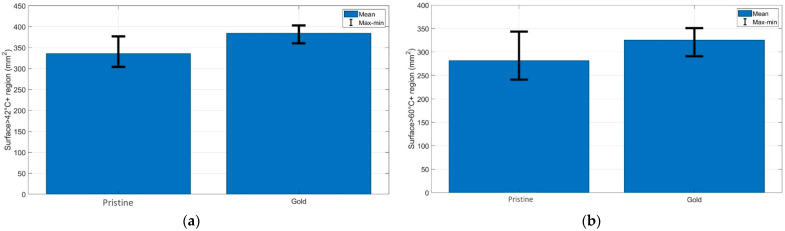
Evaluation of the surface area exposed to temperatures exceeding (**a**) 42 °C and (**b**) 60 °C at the maximum ablation temperature; the bar charts show the maximum, minimum, and mean values of the areas over three experiments for each RFA condition.

**Figure 5 biosensors-13-00008-f005:**
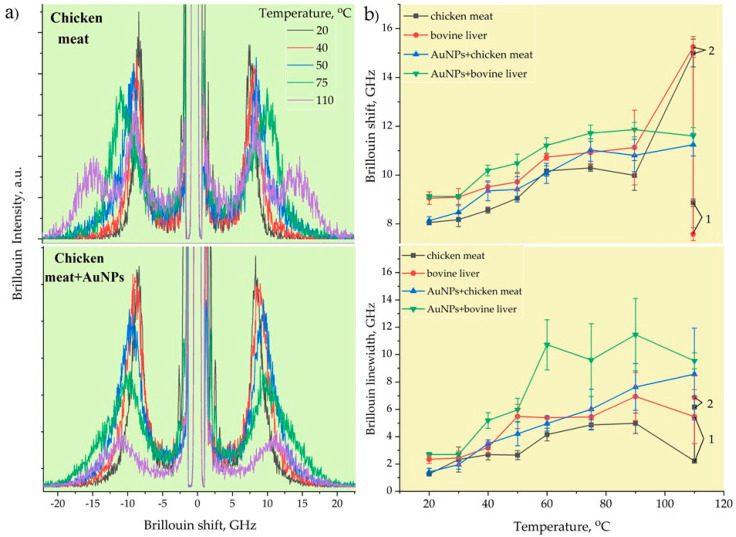
Typical Brillouin spectra of RF - ablated chicken muscle tissues without and with AuNPs (**a**). The Brillouin frequencies and linewidths of chicken meat and bovine liver with and without AuNPs (**b**).

## Data Availability

The data presented in this work are not publicly available at this time, but can be obtained upon reasonable request from the authors.

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
