# Peer review of "Thermo-Visco-Elastometry of RF-Wave-Heated and Ablated Flesh Tissues Containing Au Nanoparticles"

_biosensors, 2022, doi:10.3390/bios13010008_

Round 1

Reviewer 1 Report

This work describes the non-contact laser-based Brillouin light scattering (BLS) spectroscopy measurements of viscoelastic properties of radiofrequency (RF) hyperthermally heated and ablated bovine liver and chicken flesh tissues with embedded gold nanoparticles (AuNPs). There are some points and notes the authors should answer as pointed below.

1. In Introduction, line 130-135: The progress of Brillouin light scattering (BLS) spectroscopy measurements with the fiber optic sensors (FOS) during thermal ablation is not described. Need to improve.

2. In Figure1, how to place the fiber optic sensor. What do the fiber optic sensor realize the function? and how to realize?

3. In Figure 2 (a) and (b), SEM ruler has a problem and needs to be clarified. In Figure 2 (c) and (d), is the experimental area of the two samples consistent? Whether it is related to the type of sample in the experiment?

4. The principle is not described in the article, please improve.

5. The results in Figure 5 and 6 don’t show much performance improvement. Please explain clearly.

Reviewer 2 Report

The manuscript ID biosensors-2029888 has been devoted to mainly present an experimental and numerical study about a non-contact laser-based Brillouin light scattering spectroscopy technique employed in particular bovine liver and chicken flesh tissues with embedded gold nanoparticles. Viscoelastic properties of radiofrequency hyperthermally heated and ablated samples are reported. Please see below a list of comments to the authors:

1. How was selected the size of the Au nanoparticles for this particular application?

2. How is controlled and selected the nanoparticle density in the samples irradiated?

3. From the micrograph the nanoparticles seem anisotropic, please comment about the homogeneous or inhomogeneous response of the experiments. By the way, is there an influence of incident polarization over the main findings reported?

4. Fractional descriptions for improving the accuracy in optical and ablation effects induced in cells have been proposed. The authors are invited to discuss about this issue for future work. You can see for instance: https://doi.org/10.1016/j.ijthermalsci.2022.107734

5. A photograph of the experiment would be welcome.

6. How is the influence of the plasmonic response of the nanoparticles considering the 532 nm wavelength selected for the experiment? Is it a near-resonance or off-resonance interaction? Please justify with better details.

7. A confrontation of the main results with updated publications in the topic of this research is missing in the discussion section. You can consider for instance: https://doi.org/10.3389/fphy.2021.666192

9. It is suggested to split the collective form of the citations in order to better present the information of the selected publications to be employed in the description of the topic of this work.

10. Color bar in figure 3 is missing and the fonts in figures 2a and 2b should be enlarged.

Round 2

Reviewer 1 Report

  • Don't have any suggestions or comments

Reviewer 2 Report

The authors have improved the presentation of their work after carefully following the points raised in the review stage. In my opinion, the results are interesting and worth publication. Then I can recommend this work for this prestigious journal in present form.